# Curcumin and Its Analogs in Non-Small Cell Lung Cancer Treatment: Challenges and Expectations

**DOI:** 10.3390/biom12111636

**Published:** 2022-11-04

**Authors:** Chunyin Tang, Jieting Liu, Chunsong Yang, Jun Ma, Xuejiao Chen, Dongwen Liu, Yao Zhou, Wei Zhou, Yunzhu Lin, Xiaohuan Yuan

**Affiliations:** 1Evidence-Based Pharmacy Center, Key Laboratory of Birth Defects and Related Diseases of Women and Children, Department of Pharmacy, West China Second University Hospital, Sichuan University, Chengdu 610000, China; 2Heilongjiang Key Laboratory of Anti-Fibrosis Biotherapy, Mudanjiang Medical University, Mudanjiang 157000, China; 3Department of Pharmacy, Banan Second People’s Hospital, Banan District, Chongqing 401320, China

**Keywords:** curcumin, curcumin analogs, non-small cell lung cancer, combined treatment, signaling pathways

## Abstract

Researchers have made crucial advances in understanding the pathogenesis and therapeutics of non-small cell lung cancer (NSCLC), improving our understanding of lung tumor biology and progression. Although the survival of NSCLC patients has improved due to chemoradiotherapy, targeted therapy, and immunotherapy, overall NSCLC recovery and survival rates remain low. Thus, there is an urgent need for the continued development of novel NSCLC drugs or combination therapies with less toxicity. Although the anticancer effectiveness of curcumin (Cur) and some Cur analogs has been reported in many studies, the results of clinical trials have been inconsistent. Therefore, in this review, we collected the latest related reports about the anti-NSCLC mechanisms of Cur, its analogs, and Cur in combination with other chemotherapeutic agents via the Pubmed database (accessed on 18 June 2022). Furthermore, we speculated on the interplay of Cur and various molecular targets relevant to NSCLC with discovery studio and collected clinical trials of Cur against NSCLC to clarify the role of Cur and its analogs in NSCLC treatment. Despite their challenges, Cur/Cur analogs may serve as promising therapeutic agents or adjuvants for lung carcinoma treatment.

## 1. Introduction

Malignant lung cancer tumors are associated with the highest worldwide morbidity and mortality, with an incidence rate of 14.5 men and 8.4 women per 100,000 individuals [1]. The most common cause is long-term exposure to tobacco carcinogens that result in bronchial mucosal or gland lesions, contributing to the formation of lung tumors. Other related risk factors include environmental pollution, genetic susceptibility, chronic obstructive pneumonia, and infection [2]. The incidence of lung cancer in non-smokers is approximately 10–15 per 100,000 in the US. Notably, results of a meta-analysis indicate that lung cancer in non-smokers seems to be a distinct disease caused by mutations in driver genes, such as those for the epidermal growth factor receptor (EGFR) and anaplastic lymphoma kinase (ALK) [3]. Lung cancer has traditionally been categorized into small cell lung cancer and non-small cell lung cancer (NSCLC), the latter of which accounts for approximately 85% of all lung cancers, including lung adenocarcinoma (LUADs), lung squamous cell carcinoma (LUSCs), and large cell carcinoma subtypes [4]. However, some researchers have found that LUADs and LUSCs appear distinct in transcriptomics, pathology, and clinical treatment, suggesting that considering these two types of carcinomas as different—abandoning the existing classifications—may help develop novel specific agents [5]. At present, the clinical treatments for NSCLC include surgery, chemotherapy (such as mitomycin, gemcitabine, ifosfamide, and cisplatin), immunotherapy (including durvalumab), targeted therapy (e.g., EGFR inhibitors, vascular endothelial growth factor [VEGF] receptor inhibitors, and ALK inhibitors), and combined therapy [6]. Although these therapies have greatly improved the survival rate of patients with NSCLC, overall survival and recovery rates are still low, and tumor cells are not completely eliminated. In addition, most lung cancers originate in airway stem cell niches, indicating that lung tumors have cancer stem cells with substantial self-renewing and self-healing abilities [7]. Moreover, multiple drug resistance and cross resistance are not uncommon in NSCLC, and the etiology and pathogenesis are complex, requiring the development of more effective NSCLC treatment strategies.

Curcumin (Cur), a natural polyphenolic compound extracted from the root of turmeric, possesses diverse pharmacologic activities, including anti-diabetes [8], anti-aging [9], anti-Parkinson’s disease and Alzheimer’s disease [10], anti-cardiovascular disease [11], and anti-cancer [12], etc. Specifically, in tumor treatment, modern medicine has demonstrated that Cur exerts therapeutic effects on various cancers, including breast cancer [13], colorectal neoplasm [14,15], liver carcinoma [16,17], glioblastoma [18,19], gastric tumor [20,21], and lung carcinoma [22,23], etc. Many preclinical data have demonstrated that it is an excellent compound for different disease treatments. However, Cur also has a “dark side” exhibited in poor pharmacokinetic/pharmacodynamic (PK/PD) properties, low plasma and tissue levels, and rapid metabolism, which limits its clinical application. Researchers have attempted to change these unfavorable effects by screening Cur analogs, using piperine that interferes with glucuronidation, manufacturing liposomal Cur or nanometer Cur, and a combination of some molecule compounds [24,25]. 

In preclinical trials, Cur and its analogs have demonstrated control over NSCLC cell proliferation and metastasis, the promotion of cell apoptosis, and the regulation of autophagy via multiple mechanisms. Nevertheless, it is still unclear whether the observed Cur suppression of NSCLC in preclinical experiments occurs in humans. Hence, in this review, the objective is to clarify the role of Cur and its analogs in NSCLC treatment to accelerate the development of associated fields and potential clinical applications.

## 2. The Impact of Cur on NSCLC

We determined via a PubMed database search that researchers first reported assessing Cur as an anticancer drug in 1985 [26]. Ten years later, investigators found that Cur could suppress lung tumor metastasis and extend the life span of mice [27]. Since then, many NSCLC preclinical studies revealed that Cur could inhibit tumor nodules [27]; restrain cancer stem cells [28]; control the cell cycle [29]; suppress immigration, invasion, and repair [30,31,32]; induce the production of ROS and ER stress [33]; trigger apoptosis [34,35]; elevate DNA damage and ferroptosis [32,36]; and promote necrotic cell death [37], thereby treating and preventing NSCLC. The broad actions through which Cur can affect NSCLC in vitro and in vivo are summarized in Figure 1.

### 2.1. The Effects of Cur on the Proliferation, Invasion, and Metastasis of Lung Cancer

Malignant proliferation and highly active invasion or migration have long been considered the cause of cancer immortality. In recent years, in-depth studies of cancer progression have revealed that Cur suppresses tumors by interfering with all aspects of tumor progression, which is the action of some of the most promising anticancer drugs. First, at the root of cancer progression, Cur has been shown to elevate the ubiquitination level of TAZ that increases proteasome-degrading TAZ protein, thereby activating the hippo pathway and negatively regulating cancer stem cell function [38]. Additionally, Cur significantly impedes the self-healing of circulating cancer stem cells, limiting stem cell metastasis [28]. Cur also alters the expression of more than 700 genes linked to carcinoma development, such as those involved in DNA recovery or associated with the cell cycle, cell proliferation, or metastasis in NCI-H460 human lung cancer cells [39]. By detecting the entire transcriptome in Cur-controlled A549 cell lines, researchers revealed that Cur not only changes the expression of many genes, but also alters signaling pathways [40]. Through further investigation, it was found that those Cur-altered genes induce cell death and control extracellular matrix receptors, repressing NSCLC cell proliferation and migration [40]. These observations indicate that Cur governs NSCLC tumor growth and exhibits cytotoxic mechanisms at the genetic level.

Cur can directly stop NSCLC cell proliferation through the downregulation of Axl receptor tyrosine kinase and the inhibition of XIAP [41]. Moreover, the blockade of Bcl-2 and stimulation of Bax and cytochrome C by Cur inhibits A549 cell growth, indicating that Cur’s inhibitory action is closely related to the mitochondrial apoptosis pathway [42]. Cur also significantly hindered the angiogenesis of tumors by inhibiting the activation of STAT3 and JAK in an orthotopic xenograft model, reducing tumor size and weight [43]. Notably, heat shock protein 40 (HLJ1), a tumor suppressor, metastasis-associated protein 1 (MTA1), and E-cadherin protein play an important role in the proliferation and metastasis of NSCLC. A study on adenocarcinoma cell lines in mice showed that Cur could activate the JNK/JunD pathway to upregulate tumor suppressor HLJ1 expression and increase E-cadherin expression, blocking lung cancer cell invasion and migration [44]. It was also reported that Cur retards the expression of MTA1 along with the Wnt/β-catenin pathway to restrain proliferation and invasion in NSCLC [45,46]. Furthermore, Cur manipulates pathway crosstalk between the Wnt pathway and the adherens junction by blockading the early growth response protein (EGR-1), thereby exerting anti-proliferation and anti-metastasis effects in NSCLC 95D cells [47]. Cur also directly suppresses the levels of Toll-like receptor 4 (TLR4)/MyD88 and EGFR, thereby controlling cell cycle and epithelial–mesenchymal transition (EMT)-related checkpoints and repressing cell growth and invasion in NSCLC [29]. 

The hormone adiponectin is linked to insulin resistance and carcinogenesis. Although researchers have reported low levels of adiponectin in gastric and prostate carcinomas [48], the role of this hormone in NSCLC is controversial. Cur was shown to obstruct adiponectin receptor 1, resulting in the downregulation of adiponectin expression and the inhibition of metastasis and tumor growth of A549 cells [31]. A preclinical study of A549 cells showed that Cur blocks the migration and invasion of NSCLC by decreasing matrix metalloproteinase (MMP)-2 and 9 and VEGF; p-ERK and MEKK3 signaling pathways are also involved [49]. Notably, an investigation of MMP-9 expression and E-cadherin expression in radiation-treated A549 cells revealed markedly richer EMT than in untreated A549 cells. However, Cur inhibits the expression of MMP-9 and E-cadherin, reversing radiation-induced EMT and mitigating the invasion and metastasis of NSCLC [50].

In smokers, cigarette smoke is a potential tumor promoter because of various established carcinogens, such as polycyclic aromatic hydrocarbons, benzo(a)pyrenes, and nicotine-derived nitrosamines. These substances can alter the normal structure of lung tissue via stimulating the secretion of growth factors, neurotransmitters, and cytokines, and promote cell proliferation and cancer metastases by interfering with cell cycle progression, migration, invasion, and angiogenesis [51]. Most of these influences are due to tobacco activating cell surface neuronal nicotinic acetylcholine receptors (nAChRs) or a certain amount of β-adrenergic receptors (β-AR) to trigger downstream intracellular signaling cascades [51,52]. Considerable evidence has showed that α7-nAChR—a subtype of nAChRs—is the most powerful regulator in the oncogenic process [53,54,55]. Furthermore, in 2009, Hildegard M. Schuller explained in detail that α7-nAChR agonists can induce NSCLC cells proliferation and angiogenesis and inhibit NSCLC apoptosis through multiple pathways [56]. Therefore, nAChRs subtype-selective antagonists are regarded as a good target for the treatment of NSCLC. However, the latest research findings have revealed that Cur has been confirmed to serve as a α7-nAChR-positive allosteric modulator (PAM) [57]. Cur not only drives more Ca^2+^ entry into the cell by modulating α7-nAChR allosterically, thereby exerting neuroprotective effects [58], but also attenuates disturbed oxidative stress and improves autistic spectrum disorder by potentiating α7-nAChR [59]. Consequently, Cur may activate α7-nAChR to promote the effects of nicotine to which the lungs of smokers are exposed. If so, Cur may have detrimental effects on the treatment of NSCLC. However, there are no reports on the influence of Cur on α7-nAChR in NSCLC. In addition, due to the extensive pharmacological activities of Cur, it is possible that different doses of Cur have different effects on diverse diseases. Therefore, more research is needed to clarify the relationship between Cur and α7-nAChR to provide more evidence for the use of Cur in the treatment of NSCLC.

Nevertheless, it has also been reported that tobacco activates NF-κB, subsequently inhibiting apoptosis, promoting proliferation, and provoking tumorigenesis [60]. Cur was found to retard levels of cyclin D1, COX-2, and matrix MMP-9 to inactivate NF-κB, attenuating lung carcinogenesis in smokers [60].

### 2.2. The Effects of Cur on the Autophagy and Apoptosis of Lung Cancer

The impact of autophagy on cancers has been controversial, as autophagy can inhibit or promote tumors depending on the specific context and carcinoma progression [61]. In this way, autophagy acts dynamically in early and late-stage tumorigenesis. Cur, as an autophagy activator, causes ferroptosis, increasing the protein levels of ACSL4 and decreasing the expression of SLC7A11 and GPX4 to suppress growth and facilitate NSCLC cell death [36]. The number of NSCLC cases dramatically declines in Cur-exposed A549 cells by triggering apoptosis and autophagy through PI3K/AKT/mTOR pathway inhibition [62,63]. In addition to autophagy, Cur can induce NSCLC apoptosis by elevating the [Ca^2+^] level, resulting in Ca^2+^ overload [64]. Cur also induces oxidative stress-mediated Bcl-2 ubiquitination and Bax upregulation, causing NSCLC apoptosis [35]. The apoptosis of NCI-H460 cells is linked to an increase of ROS, intracellular Ca^2+^ and ER stress, and the FAS-caspase-8 (extrinsic) pathway in the Cur treatment group [33]. Moreover, increased ROS continually stimulates mitochondria to induce cell death and promote DNA damage, resulting in G2/M arrest and NSCLC cytotoxicity [65]. The suppression of the PI3K/AKT/mTOR pathway by Cur also provokes NSCLC apoptosis, indicating that the pathway exerts dual effects on autophagy and apoptosis [62]. Moreover, Cur also induces orthotopic NSCLC xenograft apoptosis, the mechanisms of which are associated with the suppression of the expressions of IkB, nuclear p65, COX-2, and p-ERK1/2 [66]. 

The many kinds of related signaling pathways obstructed or activated by Cur that inhibit NSCLC in vitro and in vivo are presented in Figure 2.

### 2.3. The Effects of Cur on the miRNAs and LncRNAs of Lung Cancer

Abnormal miRNA can influence cancer pathogenesis, as epigenetic modulations are considered important for carcinoma development. Natural compounds such as Cur manipulate multiple miRNAs to restore the epigenetic balance. For example, Cur upregulates miR-192-5p expression by targeting cMyc and suppressing the Wnt/β catenin pathway to limit the growth, migration, and invasion of NSCLC [67]. Furthermore, silencing miR-192-5p was found to reverse the inhibitory activity of Cur [67]. In addition to inhibiting proliferation, Cur promotes the apoptosis of H460 and A427 cells by upregulating the p53-miR-192-5p and miR-215-XIAP pathways, thereby inducing NSCLC death [34]. Similarly, Cur increases miR-206 expression by blocking the PI3K/AKT/mTOR pathway, obstructing NSCLC migration and invasion [30]. Cur has also been reported to elevate the levels of miR-330-5p (of the miR-330 family), thereby retarding A549 cell proliferation. These effects are reduced when the miR-330-5p inhibitor is added to Cur-treated A549 cells [68]. Cur was found to induce miR-98 overexpression, targeting the gene LIN28A. This activity subsequently inhibits the production of MMP-2 and MMP-9, which suppresses A549 cell migration and invasion [69]. Furthermore, the antitumor effects of Cur on NSCLC are reduced when miR-21 expression is inhibited [70]. Cur decreases circ-PRKCA levels, which is accompanied by the downregulation of ITGB1 expression and increased levels of miR-384 to repress NSCLC progression [71]. Notably, in A549/DDP multidrug-resistant human lung adenocarcinoma cells, Cur remarkably reduced the levels of miR-186 to accelerate A549/DDP cell apoptosis [72]. These findings suggest that Cur is involved in multiple crucial components of NSCLC treatment by manipulating the levels of key miRNAs (shown in Figure 3). In gemcitabine-resistant NSCLC, Cur promotes the expression of lncRNA-MEG3 and PTEN, which is linked to the inhibition of cancer growth [73]. Unfortunately, studies on Cur’s effect on NSCLC cells via interference with lncRNA expression are sparse.

## 3. The Combination of Cur and Some Small Molecular Compounds in NSCLC

Drug resistance, which is key to chemotherapy failure in tumors, occurs due to tumor heterogeneity, immune system dysfunction, microenvironment alterations, genetic influences, and other increasingly novel mechanisms [74,75]. Researchers attempt to overcome or minimize drug resistance in lung tumors to boost the efficacy of chemotherapy drugs by identifying novel mechanisms, changing therapeutic strategies, creating small-molecule therapeutic compounds or peptides, and using intensive drug combination treatments. Many studies have reported that Cur is a perfect adjunctive agent as it increases the sensitivity of NSCLC to some chemotherapeutic drugs or other anticancer agents, synergistically retarding the growth of NSCLC by regulating different mechanisms (Table 1).

### 3.1. Cur Plus an Inorganic Chemotherapy Drug

Cisplatin and carboplatin-based chemotherapy remain standard regimens in most patients with late-stage NSCLC [103]. However, recent evidence has emphasized the toxicity of and tumors’ inherent resistance to platinum-based chemotherapy, requiring researchers to consider combining platinum with other agents to minimize these adverse effects [104]. 

One study demonstrated that Cur plus cisplatin effectively inhibits the self-renewal capability of cancer stem cells (CSCs) and prevents the emergence of chemo-resistance, as these cells possess a pronounced ability to amplify, differentiate, and metastasize, resulting in cancer escape and recurrence [76]. In addition to controlling the cells’ self-renewal capacity, Cur combined with cisplatin directly induces death in the highly migratory CSC subpopulation by changing cyclin D1 and p21 expression while enhancing NSCLC sensitivity to cisplatin [79]. In addition, co-treatment with Cur and cisplatin promotes the uptake of platinum ions to suppress A549 cell survival and mediate A549 cell apoptosis by targeting the Cu-Sp1-CTR1 regulatory loop [77]. This Cur-elevated cisplatin-induced cytotoxicity in NSCLC has been linked to the downregulation of p38 MAPK-dependent X-ray repair cross-complementing group 1 (XRCC1) [78], an important mediator of DNA repair and the reparative process of cisplatin-mediated DNA injury in HepG2 cells [105]. Previous studies have indicated that XRCC1 is involved in cisplatin resistance, and the inhibition of XRCC1 contributes to facilitating DNA single-strand breaks in breast cancer cells [106,107]. In addition to XRCC1, excision repair cross-complementary 1 (ERCC1) exerts a key role in removing DNA adducts to permit the repair of damaged DNA [108]. An investigation into the potential mechanisms responsible for the anticancer effects of the co-administration of Cur and cisplatin demonstrated that combined utilization sensitizes cisplatin resistance cells to cisplatin and synergistically inhibits the proliferation of NSCLC by inactivating the ERK pathway with the subsequent suppression of ERCC1 and thymidine phosphorylase (TP) expression [80]. Based on these outcomes, the suppression of XRCC1 or ERCC1 expression by Cur combined with cisplatin provides a promising strategy for NSCLC treatment. 

Combining Cur and cisplatin has also shown that Cur intensifies apoptosis in NSCLS cells attacked by cisplatin by increasing the ROS-mediated degradation of anti-apoptotic Bcl-2 [81]. Notably, the combined use of Cur and cisplatin is also responsible for enhancing the sensitization of A549 cells to X-rays, diminishing cancer cell growth, possibly by blocking EGFR-related signaling pathways [83]. Furthermore, in cisplatin-resistant lung tumor cells, Cur plus cisplatin eliminates resistance and facilitates cytotoxic effects by blocking the FA/BRCA pathway-mediated process of DNA repair [32]. A recent study found that combining Cur, honokiol, and cisplatin increases the susceptibility of multidrug-resistant lung tumor cells to chemotherapeutics. The proposed mechanism was the inactivation of the AKT/ERK signaling pathway and the reduction of P-gp expression to limit NSCLC occurrence and progression [82]. The co-treatment of A549 cells with Cur and carboplatin revealed the suppression of the AKT-IKKα axis and downstream NF-κB and the inhibition of ERK1/2 activation, ultimately contributing to reduced cancer cell growth and elevated apoptosis [84]. Cur and palladium (II) complex combination treatment could potentially activate caspase 3/7, inducing apoptosis, thereby limiting NSCC growth and providing a new, effective method for treating lung cancer [85]. While co-treatment with Cur plus platinum or palladium could offer a promising clinical lung cancer treatment with or without platinum resistance, many challenges remain to be considered before the co-treatment can be applied.

### 3.2. Cur Plus an Organic Chemotherapy Drug

Mitomycin C, vinorelbine, gemcitabine, paclitaxel, and docetaxel are common chemotherapeutic drugs for lung cancer treatment. However, with the emergence of multiple drug-resistant tumors, clinicians are more aware of drug resistance in patients, a phenomenon that leads to the gradual weakening of a single drug’s efficacy.

Mitomycin C and Cur in combination inhibit the activation of the ERK pathway, leading to the downregulation of TP expression, which promotes DNA damage in NSCLC cells [86]. This treatment combination was also found to augment mitomycin C cytotoxicity in NSCLC by blocking the ERK pathway and suppressing the expression of Rad51, which controls tolerance in chemo- or radio-resistant neoplasms [87]. Cur combined with vinorelbine also promotes the apoptosis of NSCLC, which by mechanism was closely associated with releasing cytochrome C and activating caspase-9 along with downstream caspase-3 through the mitochondrial pathway [88].

Notably, for gemcitabine-resistant cells, the co-administration of Cur and gemcitabine greatly enhances the sensitivity of resistant cells to gemcitabine and does not appear to increase toxicity in mice [89]. Similarly, treatment with Cur and paclitaxel also sensitizes paclitaxel-resistant NSCLC cells to paclitaxel via epigenetic modification involving the upregulation of the miRNA-30c-mediated decrease of MTA1 expression [90]. Unfortunately, even though Cur plus docetaxel also synergistically retards the proliferation of NSCLC, the detailed mechanisms have not been elucidated [91].

### 3.3. Cur Plus Targeted Agents

The chemotherapy drugs crizotinib, erlotinib, and gefitinib are usually used to treat NSCLC with gene mutations. However, several exposures to crizotinib or erlotinib may result in drug resistance and therapy failure. It is reported that co-treatment with Cur and crizotinib upregulates the expression of miR-142-5p to target Ulk1 and inhibit autophagy in NSCLC cells, lowering the resistance of lung carcinoma to crizotinib [92]. Researchers also found that the co-treatment of erlotinib and Cur greatly increases the mortality of erlotinib-resistant cells due to the decreased expression of EGFR and repression of NF-κB activation in EGFR-mutant NSCLC cells [93]. Moreover, co-treatment with Cur and gefitinib significantly inactivates EGFR by retarding Sp1, influencing the interaction between Sp1 and HDAC1 and markedly promoting autophagy and autophagy-mediated apoptosis in resistant NSCLC cells [94]. Another study demonstrated that Cur and gefitinib co-treatment modulates the AKT or the p38MAPK pathway, thereby inducing apoptosis in vitro and in vivo [95]. These findings signify that Cur and targeted agents might act in concert to provide an effective therapy for advanced NSCLC.

### 3.4. Cur Plus Bioactive Molecules

When combined with Cur, other natural agents or small synthetic compounds, such as (-)-epigallocatechin gallate (EGCG), the purine analog sulfinosine, interferon-α, and galbanic acid, have been effective in NSCLC treatment. Co-treatment with Cur and EGCG, two natural agents, noticeably impedes the expression of cyclin D1 and cyclin B1, inducing NSCLC arrest at the G1 and S/G2 phases and preventing weight loss caused by tumor burden in nude mice with lung tumor xenografts [96]. Co-treatment with Cur and the natural agent galbanic in A549 cells markedly increases cancer cell apoptosis, autophagy, and other antitumor effects by suppressing the AKT/mTOR axis relative to treatment with a single chemotherapeutic agent [99]. Cur also promotes the anticancer activity of (-)-epicatechin against NSCLC by potentiating the expression of *GADD153* and *GADD45* genes, causing apoptosis and inhibiting cell proliferation [100]. In addition, Cur plus interferon-α could reverse interferon-α-induced NF-κB and COX-2 activation, suggesting that Cur counterbalances the negative effects of interferon-α and improves its antitumor activity [98]. Moreover, Cur improves the cytotoxicity of fenretinide in NSCLC by governing the expression of the ER chaperone protein GRP78 [101]. Finally, co-treatment with Cur plus multiple small molecular antagonists, such as AG1024, PD173074, LY294002, or caffeic acid phenethyl ester, also improves proliferation suppression in NSCLC by blocking the corresponding signaling pathway [102]. Although these data indicate that Cur, combined with other small agents, operates on multiple signaling pathways or integrates pathways to eliminate NSCLC, research on new compounds and combination therapies is needed to supply more clinical trial candidates.

## 4. Predicting the Interaction of Cur and Various NSCLC Molecular Targets

Discovery studio software (2019 version) was used to evaluate the interaction of Cur with amino acid residues in the binding positions of various target proteins (in 3D and 2D conformational structures, Figure 4). Various target protein (PDB) formats (www.rcsb.org, accessed on 15 August 2022) include p38MAPK [109], JAK1, JAK2, JAK3 [110,111], PI3K [112,113], GSK3-bata [114,115], TNF-alpha [116,117], and COX-2 [118]. Even though the affinity of Cur for different amino acid residues results from hydrogen, carbon–hydrogen, pi-anion, or pi-cation bonds, all target proteins can form hydrogen bonds with Cur, which disturb the activities of specific enzymes to enhance sensitivity to chemotherapy drugs or evoke apoptosis, preventing NSCLC proliferation. Preclinical and clinical studies are needed for confirmation.

## 5. The Impact of Cur Analogs on NSCLC

Cur analogs, including those that are semisynthetic and synthetic, have developed rapidly in the last 20 years. One study demonstrated that Cur exhibits anticancer effects against different types of cancer, including NSCLC in vivo and in vitro, via substituent modifications, such as dimethoxy substitution, piperidine-4-one core, or fluorination [119]. Many Cur analogs have been revealed to decrease the survival of NSCLC (Figure 5).

### 5.1. Demethoxycurcumin (DMC)

DMC, one of the most important curcuminoids, possesses antitumor activities against various human cancer cells, including leukemia, glioma cells, baller cancer, cervical cancer carcinomas, and kidney tumors [120,121,122,123,124], and is an effective adjuvant. Although few investigations have been conducted to assess the effects of DMC on NSCLC, one study on A549 cells showed that DMC downregulates TP expression and represses the expression of ERCC1. This DMC downregulation occurs by DMC acting on PI3K-AKT-snail pathways to enhance the sensitivity of NSCLC to cisplatin, thereby increasing cisplatin-induced cytotoxicity and inducing cancer cell death [125]. A recent study in A549/DDP cells revealed that DMC also markedly increases DDP sensitivity relative to DDP-resistant A549 cells. Moreover, DMC also demonstrates low toxicity to normal lung fibroblast MRC-5 cells, which indicates that the co-administration of DMC and cisplatin directly induces the apoptosis of NSCLC and enhances the anticancer effects of cisplatin for A549/DDP cells [126]. Additionally, DMC increases ROS and Ca^2+^ production and elevates the production of ER stress-related proteins, such as GRP78, GADD153, IRE1α, IRE1β, ATF-6α, ATF-6β, and caspase-4, demonstrating the beneficial effects of promoting NSCLC apoptosis [127].

However, similarly to Cur, DMC has poor water solubility, gastrointestinal absorption, and low bioavailability, which is not conducive to clinical applications [128]. Researchers have consistently focused on improving its bioavailability and changing its delivery mode into cancer cells. When cisplatin induces TP and ERCC1 overexpression, DMC-loaded amphiphilic chitosan nanomatrix (DMC–CHC) significantly enhances cisplatin-mediated cytotoxicity via the high-performance intracellular transmission of the encapsulated DMC. Moreover, DMC–CHC combined with cisplatin greatly suppresses cisplatin resistance protein ERCC1 and TP through the PI3K-AKT pathway, permitting increased cisplatin-induced NSCLC apoptosis [129]. Notably, DMC–CHC covered with an anti-EGFR antibody layer to assist DMC entrance into the cytoplasmic region exhibits high cytotoxicity against multidrug-resistant lung cancer, particularly NSCLC [130]. This antibody shell layer is accurately positioned on the cell against tumor cells and slows down drug release. This technique is valuable for developing drug delivery systems for NSCLC treatment.

### 5.2. Bisdemethoxycurcumin (BDMC)

BDMC comprises approximately 3% of curcuminoids and is more stable than others in vivo [131]. Recent studies have suggested that BDMC plays an antitumor role via multiple biological mechanisms or synergistic modes of action, from the suppression of cell growth, proliferation, invasion, and migration, to the induction of programmed cell death in various cancers [132,133,134,135,136,137].

First, BDMC significantly induces DNA lesions and restrains DNA repair by inhibiting the expression of the proteins 14-3-3σ, MGMT, BRCA1, and MDC1 and increasing p-p53 and p-H2A.X in NSCLC [138]. BDMC also alters the expression of specific genes, including *CCNE2*, linked to the cell cycle, and *ERCC6L*, involved in DNA damage and recovery, which mediates the cytotoxicity, disturbing cell migration, invasion, and tumor progression for NSCLC treatment [139]. In addition to affecting the levels of genes, BDMC can block the activity of DNA methyltransferase-1. Blocking this DNA activity decreases Wnt inhibitory factor-1 (WIF-1) promoter demethylation associated with the activation of the Wnt pathway, resulting in epigenetic alterations that inhibit the Wnt pathway and induce apoptosis in NSCLC [140]. Moreover, the Wnt pathway also participates in the progression of EMT and cancer metastasis [141]. When the Wnt pathway is activated, secreted Wnt proteins operate on E-cadherin to promote the EMT [142]. BDMC negatively regulates the Wnt/β-catenin pathway by elevating WIF-1 expression, which subsequently blocks TGF-β1-induced EMT to inhibit migration and invasion in highly metastatic NSCLC [143]. In addition, BDMC causes autophagy, upregulating the levels of E-cadherin and downregulating vimentin to limit the migration and invasion of highly metastatic NSCLC [144].

Accordingly, the BDMC induction of autophagy accelerates cell apoptosis via the inactivation of the hedgehog signaling pathway in NSCLC [145]. Moreover, BDMC strongly induces S-phase arrest by suppressing the protein levels of Cdc25A and cyclins A and E, promoting DNA damage by increasing ROS and Ca^2+^ production and raising ER stress by altering the protein expression of GRP78, GADD153, IRE1α, IRE1β, ATF-6α, ATF-6β, and caspase-4 expression to induce lung tumor cells apoptosis [146].

Two studies have demonstrated that BDMC enhances the sensitivity of NSCLC to chemotherapeutic drugs. BDMC combined with icotinib exhibits potent, synergistic growth suppression in EGFR-TKI-resistant NSCLC via multiple pathways, such as ROS accumulation, autophagy induction, or DNA damage [147]. Furthermore, there may be crosstalk between several signaling pathways because single-target agents, such as ROS antagonists (n-acetylcysteine) or autophagy antagonists (chloroquine or 3-MA), only partially abate growth inhibition induced by BDMC plus icotinib in NSCLC cells [147]. Furthermore, BDMC and cisplatin co-treatment regulates the chemo-sensitivity response for cisplatin-resistant NSCLC by repressing the expression of CA916798 and PI3K/AKT activities [148]. CA916798 is a recently identified protein that interacts with AKT in BDMC-exposed A549 and A549/CDDP cells. Moreover, the phosphorylation of CA916798 at the S20 residue exerts a key role in mediating BDMC anti-carcinoma activity [148]. All these data demonstrate that BDMC inhibits NSCLC cell growth, proliferation, invasion, and migration by extrinsic, intrinsic, and ER stress pathways to drive cell apoptosis in NSCLC.

### 5.3. Cur Analogs with Targeted Therapy

Oncogenic receptor tyrosine kinases are definitive drug therapy targets for the treatment of NSCLC patients containing gene mutations, especially EGFR mutations and ALK rearrangements that are targets of the precision medicine management of chest neoplasms [149,150]. A meta-data analysis showed that individuals who never smoked were more susceptible to the EGFR and ALK-EML4 mutations than ever-smokers [3]. In ALK+ cells, multiple effective ALK tyrosine kinase inhibitors (TKIs), such as crizotinib, ceritinib, alectinib, brigatinib, or lorlatinib, contribute to substantial improvement [151]. However, with increased adverse effects and the appearance of molecular resistance, the identification of new TKIs for NSCLC targets or combinations is urgent.

Two Cur analogs, RL66 and RL118, showed powerful anticancer activities against various tumor cells [152,153]. An investigation into ALK+ (H3122) and ALK− (A549) cells revealed that RL66 and RL118 demonstrate greater potency in ALK+ cells than in ALK− A549 cells. Moreover, co-treatment with crizotinib and RL66 or RL118 promotes ALK+ H3122 cell death more than either drug alone [136]. Moreover, in crizotinib-resistant H3122 cells, RL66 and RL118 demonstrate negligible cross resistance with crizotinib, indicating that RL66 or RL118 act on independent targets, requiring further exploration in vivo [154]. 

In NSCLC patients with EGFR mutations, EGFR-TKIs such as gefitinib and erlotinib are the first-line clinical drugs, dramatically improving quality of life. However, most of these patients obtain resistance to EGFR-TKIs via a secondary mutation within EGFR [155,156]. However, Cur analog WZ35 suppresses the proliferation of gefitinib- and erlotinib-resistant H1975 cancers via increased ROS generation, accompanied by ER stress and mitochondrial disorder, which trigger apoptosis [157]. Cur analog CUCM-36 targets EGFR to prevent NSCLC via diverse mechanisms, whereas the actual synthesis of the CUCM-36 compound is required to verify its anti-EFGR action in vivo and in vitro [158]. These data offer insight to accelerate studies on Cur analog-targeted therapy for NSCLC patients with gene mutations.

### 5.4. Other Cur Analogs

CA-5f, a Cur analog that is well tolerated in vivo, is toxic to NSCLC via elevated ROS production in mitochondria and diminishes the growth of NSCLC by blocking autophagosome–lysosome fusion [159]. Similarly, Cur analog EF24 also demonstrates excellent cytotoxicity against NSCLC cells by increasing ROS deposition and subsequent mitochondrial disorder-mediated apoptosis to repress tumors [160]. However, in contrast to CA-5f, EF24 induces autophagy to prevent the survival of NSCLC rather than inhibit autophagy. Another Cur analog, MS13, demonstrates potential anticancer activities that inhibit most carcinomas, such as colon cancer, prostate tumor, glioblastoma, cervical carcinoma, and neuroblastoma [161]. Studies exploring the influence of MS13 on NCI-H520 and NCI-H23 lung cancer cells showed that MS13 promotes anti-proliferation and apoptosis activity in NSCLC, involving the PI3K-AKT axis, cell cycle-apoptosis, and MAPK signaling pathways [161]. Unfortunately, researchers have not fully elucidated the mechanisms of MS13 against NSCLC, so further analysis is needed. With a triazole ring, the analog compound 5k regulates multiple signaling pathways, including MAPK activation and STAT3 and NF-κB inhibition, to reduce lung tumor development [162]. Although PI3K-AKT pathways play a significant role in tumor growth, invasion, migration, and apoptosis, 5k does not influence PI3K-AKT pathways [162].

ER stress has recently been regarded as a significant driver of cell apoptosis [163], so disrupting ER functions to elevate ER stress may be a treatment for various cancers. One investigation indicated that Cur analog B86 diminishes cancer growth by inducing ER stress-mediated apoptosis, whereas silencing the CHOP gene reverses the inhibitory effects of B86 in NSCLC [164]. Similarly, Cur analog B63-treated NSCLC has been shown to cause lung tumor cell apoptosis by regulating ER Ca^2+^ stores and stimulating the production of ER stress, ultimately triggering the caspase cascades [165]. 

Cur derivative MHMM-41 appears to be a more promising agent against NSCLC in vitro. MHMM-41 increases ROS production, alters mitochondrial permeability or mitochondrial membrane potential, and activates caspase-3, 8, 9, 12, Bax, and PARP proteins to cause A549 cell apoptosis. These results suggest that MHMM-41 exerts inhibitory effects on NSCLC through extrinsic and intrinsic mitochondrial pathways [166]. Similarly, Cur analog A501 also promotes the apoptosis of NSCLC cells by inhibiting cyclinB1, cdc-2, and Bcl-2 and activating p53 and caspase-3 [149]. Analog A501 induces G2/M arrest by interfering with cyclin expression to limit cancer development in NSCLC, as cyclins play a vital role in cell proliferation, irreplaceable for cell cycle transitions [167]. In another study, investigators synthesized Cur analog A17 (with a double carbonyl group) and explored its antitumor effects against NSCLC. The data revealed that A17 exerts potential anticancer activity by increasing CHOP expression and some crucial constituents to induce ER stress. Moreover, this research reported that A17 induces H460 cell apoptosis by successively increasing the production of downstream proteins, such as p53, p-JNK, and Bax, ultimately activating caspase-3 to induce NSCLC apoptosis [168]. 

Two Cur derivatives, MOMI-1 and HBC, repress the proliferation of A549 cells by causing autophagy rather than cell apoptosis. Autophagy has been regarded as a crucial intracellular mechanism for the degradation of damaged proteins and organelles, while the activation of autophagy can suppress NSCLC progression [169]. Microtubule-associated protein LC3 is an important marker for estimating whether autophagy is activated. When autophagy occurs, the LC3-I is converted to LC3 II rapidly, increasing the LC3-II/I ratio. However, LC3-I/II conversion is not the only measure used to judge autophagy. MOMI-1 not only promotes the conversion of LC3-I to LC II to induce autophagy, but also inhibits the migration of A549 cells and disturbs the cell cycle [170]. By contrast, HBC promotes the fusion of autophagosomes with lysosomes and increases the conversion of LC3-I to LC3 II to induce autophagy, ultimately repressing A549 proliferation [171]. Another Cur derivative, ZYX01, also upregulates the ratio of LC3-II/LC3-I and beclin-1 and reduces p62 expression to provoke autophagy [172]. Mechanistic studies show that ZYX01 activates the AMPK pathway, which further facilitates ULK1 phosphorylation to inhibit A549 cells, while using autophagy inhibitors eliminates the effect of ZYX01 on the AMPK/ULK1/beclin-1 pathway and autophagy [172]. Notably, tetrahydrocurcumin increases the formation of autophagosomes by elevating beclin 1 expression and advancing autophagic cell death by blocking the PI3K/AKT/mTOR pathway, inducing autophagy [173]. In the early stages after tetrahydrocurcumin administration into A549 cell cultures, the LC3-II/LC3-I ratio is low. The ratio increases over time [173]. These findings demonstrate that the activation of the AMPK/ULK1/beclin-1 pathway or the inhibition of the PI3K/AKT/mTOR pathway promotes the autophagy of NSCLC cells. However, studies have revealed that autophagy activation prompts several types of cancer cells to obtain the energy and metabolites required to maintain their growth and decreases the sensitivity of NSCLC to chemo/radio-therapeutic drugs [174,175,176,177,178]. Preclinical and clinical research has shown that the inhibition of autophagy could attenuate tumor development or improve chemo/radio-therapy resistance in NSCLC [115,179]. Notably, Cur analog CB-2 increased the accumulation of LC3B-II, yet this did not change other key proteins related to autophagy. CB-2 also reduces tumor growth by inhibiting autophagosome–lysosome fusion [180]. The results of additional studies suggest that lower-dose CB-2 represses the migration of A549 cells while slightly affecting apoptosis, whereas higher doses of CB-2 directly promote apoptosis and necrosis in A549 cells [180]. Therefore, autophagy inhibitors have also emerged as efficient NSCLC antitumor agents. It is unclear why both autophagy inducers and inhibitors provoke the apoptosis of NSCLC. The interactions between autophagy and apoptosis are intricate, and autophagy acts as a double-edged sword in NSCLC treatment [181]. 

For cisplatin-resistant A549 cells, Cur analog 2a effectively enhances the sensitivity of A549/CDDP cells to cisplatin by increasing oxidative stress. This is because 2a, as a thioredoxin reductase (TrxR) inhibitor, can deplete glutathione and disrupt intracellular redox homeostasis, thereby restraining cell growth [182]. TrxR with selenocysteine exhibits strong anti-oxidant stress, possibly regulating redox balance to influence the development of cancers by mediating signal transduction [183,184]. Moreover, the suppression of TrxR augments the radio-sensitization of cancer to accelerate cell apoptosis [185]. Hence, using TrxR inhibitors or the co-administration of TrxR suppressants and chemoradiotherapy drugs provides a new NSCLC treatment methodology [186,187,188]. Another study revealed that Cur analog MHMD regulates cell death by apoptosis and necrosis pathways, although the detailed regulation mechanisms are still unclear [189]. Notably, however, MHMD activates caspases to induce apoptosis rather than autophagy, inhibiting NSCLC cell growth [189]. Unfortunately, these studies suffer from a lack of research in animal samples, and data on the inhibition of NSCLC require corroboration via larger preclinical studies. The associated analogs are listed in Table 2.

## 6. Clinical Trials of Cur and Its Analogs in NSCLC

In a phase I clinical trial (NCT02321293), the aim of co-administration of Cur and TKIs for EGFR-mutant-advanced NSCLC was to assess the safety and feasibility of using Cur in conjunction with an EGFR-TKI in patients with advanced NSCLC. The 20 study subjects (≥18 years) were given CURCUVivaTM (an enhanced bioavailable formulation of Cur) at 80 mg per day (in capsule form) for eight weeks, after which the patients continued taking EGFR-TKI (gefitinib, 250 mg/once daily or erlotinib, 150 mg/once daily) without Cur until progression.

Researchers also investigated (JPRN-UMIN000006892) the safety and feasibility of erlotinib and theracurmin (nanoparticle Cur) co-treatment in patients with advanced or recurrent NSCLC. Patients were treated with theracurmin and erlotinib for eight weeks (erlotinib, 150 mg per day; theracurmin, three cohorts: 180 mg per day, 240 mg per day, and 360 mg per day). In another phase I/Π clinical trial (NCT01048983), researchers compared armodafinil, bupropion, Cur, and minocycline alone or in combination to determine the most appropriate treatment for controlling symptoms of lung cancer. Unfortunately, this project was withdrawn. In the latest phase Π clinical trial (NCT03598309), researchers attempted to determine whether co-treatment with Lovaza (made with fish oils) and Cur C3 complex was nontoxic and tolerable and could help to reduce the size of lung nodules. After searching multiple clinical research websites, including https://www.isrctn.com/ (accessed on 3 July 2022), https://www.anzctr.org.au/ (accessed on 3 July 2022), https://www.clinicaltrialsregister.eu/ (accessed on 3 July 2022), https://irct.ir/page/about_irct (accessed on 3 July 2022), http://ctri.nic.in/Clinicaltrials/login.php (accessed on 3 July 2022), and http://apps.who.int/trialsearch/ (accessed on 3 July 2022), we identified no current studies on Cur and its analogs for NSCLC treatment.

## 7. Conclusions

Considerable preclinical evidence has revealed that Cur and its analogs affect NSCLC via various mechanisms, such as inducing ROS production, increasing ferroptosis, changing mitochondrial potential, and disturbing cellular signaling pathways. Additionally, co-treatment with Cur and other agents synergistically enhances cytotoxicity in NSCLC cells to suppress tumor cell growth, migration, and invasion. This evidence suggests that Cur and its analogs offer promise to prevent NSCLC in humans. However, substantial clinical research is required before Cur, Cur analog, and Cur co-treatment with other chemotherapeutic agents can be implemented as anti-lung cancer drugs in clinical settings. First, some results have been less than encouraging. Second, critical issues remain regarding dosage, formulations and bioavailability, optimal combinations, potential adverse reactions, and other factors; thus, more studies are required.

## Figures and Tables

**Figure 1 biomolecules-12-01636-f001:**
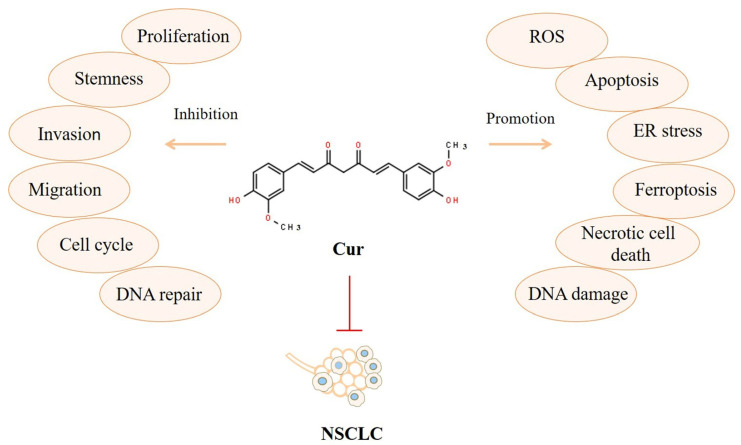
Broad actions of Cur against NSCLC.

**Figure 2 biomolecules-12-01636-f002:**
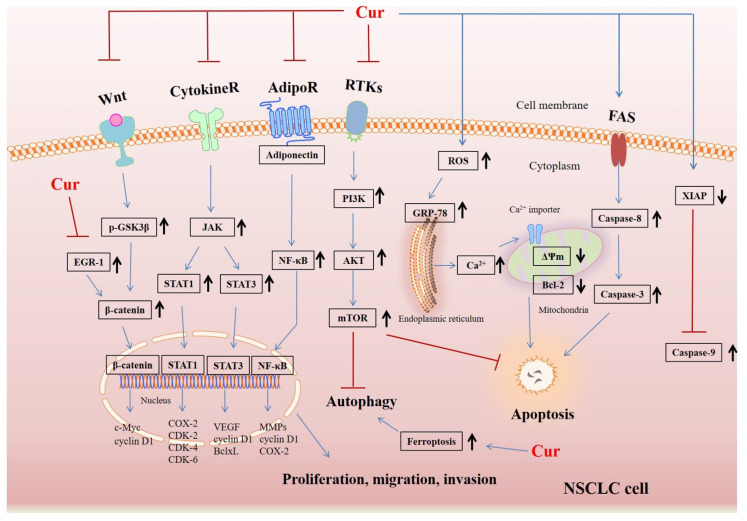
The associated signaling pathways through which Cur suppresses NSCLC.

**Figure 3 biomolecules-12-01636-f003:**
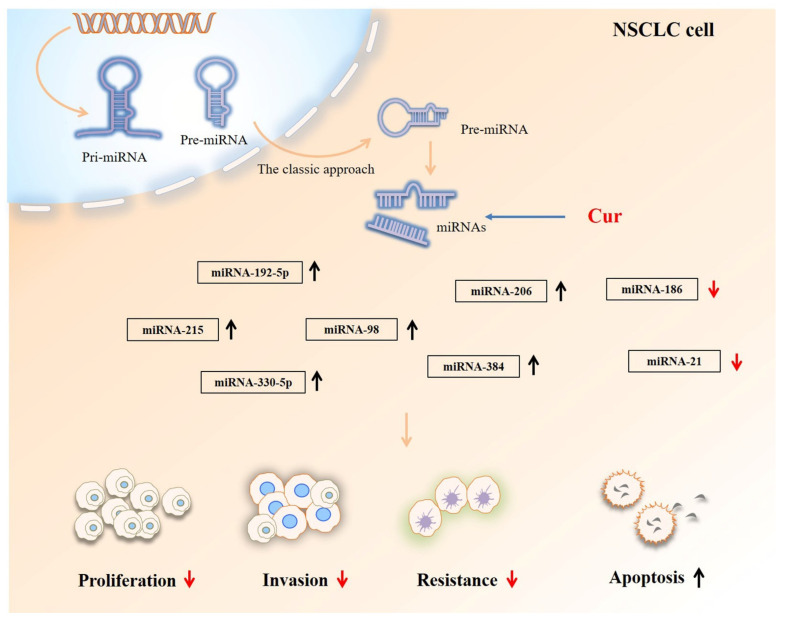
Cur controls the levels of specific key miRNAs against NSCLC.

**Figure 4 biomolecules-12-01636-f004:**
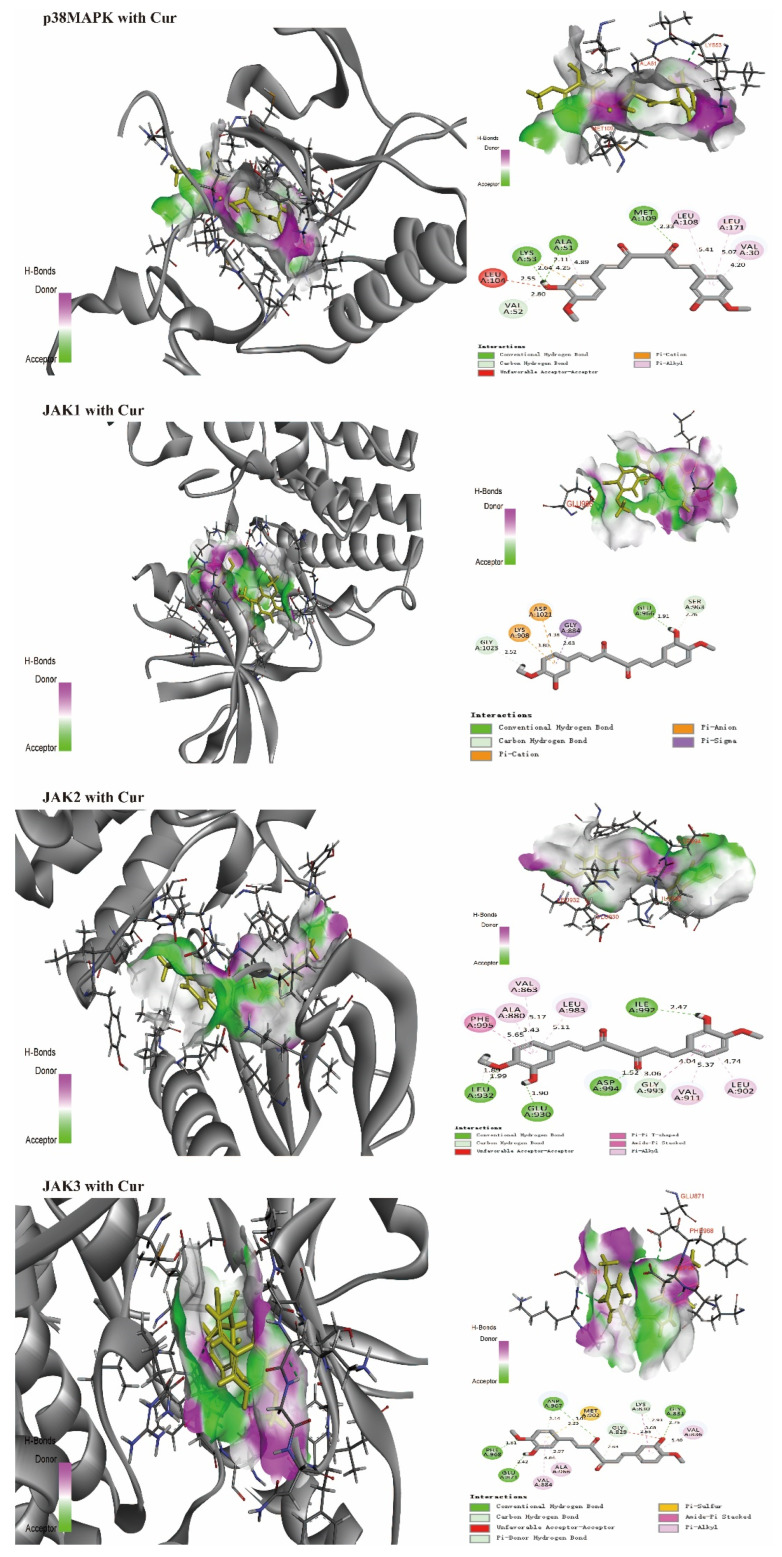
Interactions (3D and 2D) of Cur with various amino acid residues in NSCLC-related proteins. p38MAPK (hydrogen bonds: ALA51, LYS53, and MET109), PDB: 5WJJ; JAK1 (hydrogen bond: GLU966), PDB: 6SMB; JAK2 (hydrogen bonds: GLU930, LEU932, ILE992, and ASP994), PDB: 3UGC; JAK3 (hydrogen bonds: GLY831, GLU871, ASP967, and PHE968), PDB: 3ZC6; PI3K (hydrogen bonds: GLU880, VAL882, ASP950, and ASN951), PDB: 3TL5; GSK3-bata (hydrogen bonds: TYR134, PRO136, and ARG141), PDB: 7OY5; TNF-alpha (hydrogen bonds: GLN61, GLY121, and TYR151), PDB: 2AZ5; COX-2 (hydrogen bonds: MET522 and ALA527), PDB: 4M10.

**Figure 5 biomolecules-12-01636-f005:**
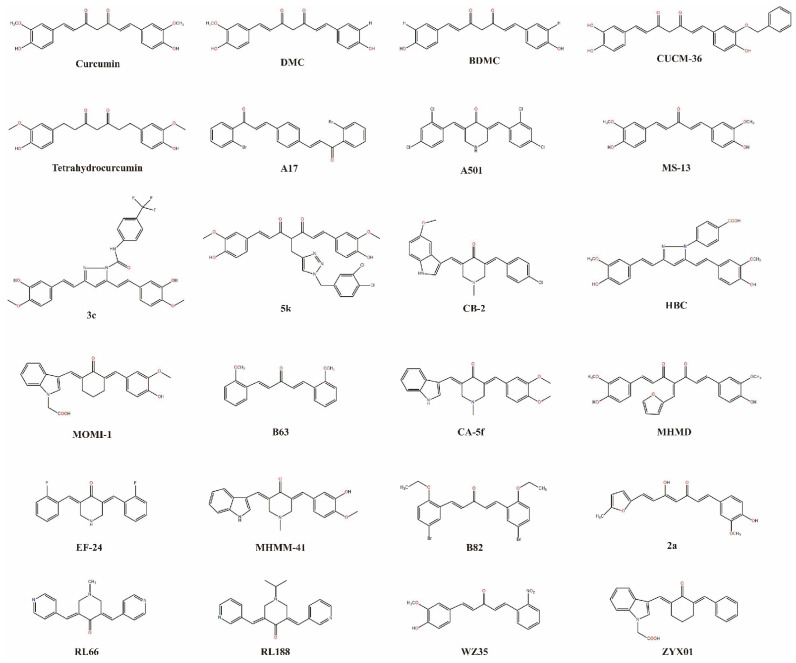
Various Cur analogs used against NSCLC.

**Table 1 biomolecules-12-01636-t001:** Combination of Cur and some small molecular compounds against NSCLC in preclinical studies.

Items	Dosage	Assay Type	Mechanisms	Outcomes	References
Cur plus inorganic chemotherapy drug					
Cur plus cisplatin	41 μM Cur and 30 μM cisplatin for A549 cells; 33 μM Cur and 7 μM cisplatin for H2170 cells	A549 and H2170 cells	Suppression of the self-renewal capability of cancer stem cells	Synergistic inhibition of NSCLC	[76]
Cur plus cisplatin	In vitro: 2–32 µM Cur and 0.5–8 µg/mL cisplatin In vivo: 2.5 mg/kg cisplatin and 50 mg/kg Cur	A549, H1299, and NCI-H460 cells and BALB/c mice	Upregulating the levels of CTR1 and Sp1 to increase more Pt^2+^ uptake	Enhancing sensitivity and antitumor effects of cisplatin in NSCLC	[77]
Cur plus cisplatin	40 μM Cur and 20 μM cisplatin	A549 and H1703 cells	Decreasing XRCC1 expression to restrain the repair of platinum-DNA in NSCLC cells.	Enhancing cytotoxic effect in NSCLC cells	[78]
Cur plus cisplatin	10–40 µM Cur and ≤3 µM cisplatin	A549 and H2170 cells	Triggering intrinsic apoptotic pathway to limit CSC growth	Elevating cisplatin sensitivity of the double-positive (CD166+/EpCAM+) CSC subpopulation in NSCLC cells	[79]
Cur plus cisplatin	0–20 µg/mL Cur and 0–20 µg/mL cisplatin	A549 and A549/DDP cells	Downregulation of FA/BRCA pathway DNA damage repair processes to induce apoptosis in NSCLC	Sensitizing cisplatin-resistant NSCLC cells to cisplatin	[32]
Cur plus cisplatin	0–40 µM Cur and 0–20 µg/mL cisplatin	A549 and H1975 cells	Inactivating ERK pathway to decrease expression of TP and ERCC1	Synergistic suppression of NSCLC	[80]
Cur plus cisplatin	0–100 μmol/L Cur and 0–100 μmol/L cisplatin	NCI-H460 cells	Raising superoxide anion generation to diminish Bcl-2 protein	Enhancing cytotoxic effect of cisplatin in NSCLC cells to induce apoptosis	[81]
Cur plus honokiol (HNK) plus cisplatin	10 µg/mL Cur and 5 µg/mL cisplatin and 5 µg/mL HNK	A549 and A549/DDP cells	Downregulation of P-gp expression and inactivating AKT/ERK pathway to promote apoptosis and suppress migration and invasion	Synergistically elevating sensitivity of multidrug-resistant NSCLC to cisplatin	[82]
Cur plus cisplatin plus X-ray	10 µmol/L Cur and 1 mg/L cisplatin	A549 cells	Associated with blocking the EGFR-related signaling pathway.	Augmenting radiosensitization effects against NSCLC	[83]
Cur plus carboplatin	10 µM Cur and 50 or 100 µM carboplatin	A549 cells	Activation of ERK1/2 and suppression of Akt/IKKα pathway to inhibit NF-Κb	Synergistically promoting apoptosis and suppressing tumor cell growth, migration, and invasion in NSCLC	[84]
Cur plus palladium complex	0.78–100 μM Cur and 0.39–50 μM Pd (II) complex	H1299 and A549 cells	Activation of d caspase 3/7 to induce apoptosis	Exhibiting a superior cytotoxic activity to suppress tumor growth	[85]
Cur plus organic chemotherapy drug					
Cur plus mitomycin C (MMC)	5–50 µM Cur and 0.5–5 µM MMC	H1975 and H1650 cells	Downregulating TP expression and inactivating ERK1/2 pathway	Synergistic increasement of MMC-induced cytotoxicity	[86]
Cur plus MMC	0–40 µM Cur and 0–10 µM MMC	A549 and H1975 cells	Decreasing the expression of Rad51 and blocking the MKK1/2–ERK1/2 pathway	Synergistic increasement of MMC-induced cytotoxicity	[87]
Cur plus vinorelbine	25 μM Cur and 0.1 μg/mL vinorelbine	NCI-H520 cells	Releasing cytochrome C and activating caspase-9 and downstream caspase-3 in mitochondria to promote apoptosis	Synergistic inhibition of NSCLC	[88]
Cur plus gemcitabine (GEM)	3 μmol/L Cur and 58.2 μmol/L GEM for A549 cells; 3 μmol/L Cur and 98.72 μmol/L GEM for A549/GEM cells	A549 and A549/GEM drug-resistant cells	Downregulating expression of MMP-9, vimentin, and N-cadherin and upregulating E-cadherin to slow EMT	Elevating sensitivity of GEM-resistant NSCLC and decreasing migration and invasion	[89]
Cur plus paclitaxel	30 µM Cur and 30 µM paclitaxel	A549 and H460 cells and paclitaxel-resistant lines A549 and H460	Upregulation of miR-30c-5p to decrease levels of MTA1	Increasing paclitaxel sensitivity to paclitaxel resistant NSCLC cells	[90]
Cur plus docetaxel (Doc)	In vitro: 2 µM Cur and 2 nM Doc or 0.5 µM Cur and 0.5 nM Doc In vivo: 15 mg/kg Cur and 10 mg/kg Doc	A549 cells and nude mice	Not clarified	Synergistic suppression of NSCLC	[91]
Cur plus targeted agents					
Cur plus crizotinib	30 μM Cur and 20 μM crizotinib	A549, H460, H1299, and H1066 cells	Increasing the levels of miR-142-5p through epigenetic and suppressing autophagy	Enhancing NSCLC sensitivity to crizotinib treatment	[92]
Cur plus erlotinib	12.5 µM Cur and 1 µM erlotinib	H1650 and H1975 cells	The blockade of NF-κB activation and reducing the expressions of EGFR, p-EGFR, and survivin	Lowering erlotinib-resistant NSCLC cells with mutated EGFR	[93]
Cur plus gefitinib (Gef)	In vitro: 5–10 μM Cur and 0–20 μM Gef In vivo: 1 g/kg Cur and 100 mg/kg Gef	H157, H1299, and PC-9 cells and BALBL/c mice	Inhibition of Sp1/EGFR activity to induce autophagy-mediated apoptosis	Elevating the sensitivity to Gef in NSCLC patients with mutated EGFR	[94]
Cur plus Gef	In vitro: 1–20 µM Cur and 1–20 µM Gef In vivo: 1 g/kg Cur and 120 mg/kg Gef	CL1-5, A549, H1299, H1650, and H1975 cells and SCID mice	Reducing the levels of EGFR and altering p38MAPK or inhibiting AKT to promote apoptosis	Improving the treatment in NSCLC patients with mutated EGFR	[95]
Cur plus bioactive molecules					
Cur plus (-)-epigallocatechin gallate (EGCG)	In vitro: 10 μM/L Cur, 10 μM/L EGCGIn vivo: (20 mg/kg) EGCG and Cur	A549 and NCI-H460 cells and BALB/c nude mice	Suppression of proteins cyclin D1 and cyclin B1 to enhance cell cycle arrest	Synergistic inhibition of NSCLC	[96]
Cur plus the purine analog sulfinosine (SF)	7.5 or 35 µM Cur and 1–10 µM SF for NCI-H460; 15 or 55 µM Cur and 5–25 µM SF for NCI-H460/R	NCI-H460 and NCI-H460/R cells	Linked with cell cycle distribution	Synergistic inhibition of multidrug-resistant NSCLC	[97]
Cur plus interferon-α (IFN-α)	0–50 µM Cur and 1000U/mL IFN-α	A549 cells	Inhibition of COX-2 and NF-κB activation	Overcoming the resistance of NSCLC to IFN-α	[98]
Cur plus galbanic acid (GBA)	10–20 μM Cur and 40 μM GBA	A549 cells	Suppression of Akt/mTOR pathway	Potentiating the antitumor effect of GBA and inhibiting cancer migration	[99]
Cur plus (-)-epicatechin (EC)	15–25 μmol/L Cur and 100 or 200 μmol/L EC	PC-9 and A549 cells	Upregulating the levels of GADD153 and GADD45	Remarkable improvement in growth inhibition and apoptosis of NSCLC	[100]
Cur plus fenretinide (Fen)	In vitro: 10–20 μM Cur and 4–6 μM Fen In vivo: 40 mg/kg Cur and 1 mg/kg Fen	A549 and LLC cells and C57BL/6 mice	Related to regulating ER chaperone protein GRP78	Synergistic suppression of NSCLC	[101]
Cur plus multiple small molecular agents	5–10 µM Cur and 0.1–2.5 µM AG1478, AG1024, PD173074, LY294002, or caffeic acid phenethyl ester (CAPE)	H1299 and A549 cells	Associated with EGFR, insulin-like growth factor 1, fibroblast growth factors receptor, PI3K, or NF-κB signaling pathway	Cooperating with these small molecules to inhibit tumor growth	[102]

**Table 2 biomolecules-12-01636-t002:** Various Cur analogs and their mechanisms of action against NSCLC.

Analogs	Dosage	Assay Type	Mechanisms	Outcomes	References
CA-5f	0–40 µM	A549 cells and BALB/c nude mice	Increasing ROS production in mitochondria and inhibiting autophagy	Suppressing the growth of NSCLC	[159]
EF24	2 μM, 4 μM/ 5, 10 or 20 mg/kg	A549 and H520 cells and BALB/c nude mice	Increasing ROS production in mitochondria, autophagy, and apoptosis	Inhibiting the growth of NSCLC	[160]
MS13	6.3 µM, 12.5 µM	NCI-H520 and NCI-H23 cells	Greatly concerned with PI3K-AKT signaling pathways, cell cycle-apoptosis, and MAPK pathways	Inducing anti-proliferation and apoptosis activities of NSCLC	[161]
MHMM-41	8 μM and 16 μM	A549 cells	Activating caspase-3, 8, 9, 12, Bax, and PARP proteins and increasing the levels of ROS	Inducing apoptosis of cancer cells and inhibiting tumor migration	[166]
A17	1, 5, 10 µM	NCI-H460 and A549 cells	Activating PERK, ATF-6, and IRE1 to increase ER stress	ER stress-induced cell apoptosis	[168]
A501	0.5, 1, 2, 4 µM	A549, H460, H1975, and HCC827 cells	Inhibiting the levels of cyclinB1, cdc-2, and Bcl-2 and increasing p53, cleaved caspase-3, and Bax	Inhibition of cancer cell proliferation and causing apoptosis of NSCLC cells.	[167]
B82	2.5, 5, 10 µM and 5 mg/kg/day	H460 cells and BALB/c nude mice	Inhibiting the levels of cyclinB1 and Bcl-2 and increased cleaved caspase-3 and Bax	Inhibition of cancer cell proliferation and causing apoptosis of NSCLC	[164]
B63	10 and 20 μM	H460 and H358 cells and BALB/c nude mice	Regulating intracellular Ca^2+^ in ER to increase cleaved caspase-3 and cleaved caspase-9	ER stress-induced cell apoptosis	[165]
5 k	0.625 μM–2.5 μM	A549 cells and zebrafish larvae	Activation of the mitogen-activated protein kinases and suppression of NF-κB/STAT3 signaling pathways	Inhibition of cancer cell proliferation	[162]
MOMI-1	5, 20, 80 μM	A549 cells	Through the LC3-I/II conversion and reducing the levels of cyclin D1 and cyclin E1 protein	Inhibition of cancer cell proliferation and migration	[170]
HBC	0–80 μM	A549 cells	Increasing amassment of autophagosomes and inducing the conversion of LC3-I to LC3-II	Inducing cell autophagy to suppress tumor proliferation	[171]
MHMD	0–20 µM	A549 cells	Unclearly specific mechanisms	Inducing cell necrosis and death	[189]
ZYX01	15 μM	A549 cells	Activation of AMPK- ULK1-Beclin-1 signaling pathway	Inducing autophagy and inhibiting migration in NSCLC cells	[172]
CB-2	0–40 μM	A549 cells	Elevating ROS production in mitochondria and inhibiting autophagy	Suppressing the growth and migration of NSCLC cells	[180]
2a	0–100 μM	A549 and A549/CDDP cells	Depleting GSH to promote oxidative stress damage and apoptosis	Enhancing drug sensibility to cisplatin against A549/CDDP cells	[182]

## Data Availability

Not applicable.

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
