# Peer review of "Curcumin and Its Analogs in Non-Small Cell Lung Cancer Treatment: Challenges and Expectations"

_biomolecules, 2022, doi:10.3390/biom12111636_

Round 1

Reviewer 1 Report

Thank you for the opportunity to review this manuscipt.

The topic is very interesting but I think it's necessary to better specify the goal of this manuscrupit. In the introduction, the issue should be clear. In particular in this case, I don't undestand what is the aim of the study. The objective was to clarify the role of curcumina and its analogs in non-small cell lung cancer treatment? Please added this sentence. In addition, in the introduction I think that the interest in this substance should be better specified. What are its properties? Are there other studies that investigated its role in cancer treatment?

The research methodology is lacking.What's this about?Is this a narrative review? How was the information provided selected?What research database were consulted? All are extremely important information to provide to the reader.

Reviewer 2 Report

This is a very extensive review on curcumin its analogs and effects on NSCLC lung cancer cells, signalling pathways and elements involved in such pathways. I have only few remarks as far integrity of the manuscript is concern. It is known that NSCLC express various types of nAChR, especially alpha7 sub type and there are many reports on involvement on these receptors in cancer proliferation, inhibition of apoptosis and enhancing angiogenesis. It is known, though there is some controversy in these reports, that agonists of nAChR triggers signalling pathways that lead to apoptosis, prevent proliferation and/or angiogenesis and therefore countereffect of nicotine to which lungs are exposed in smokers  There are recent reports that curcumine modify the activity of these receptors mainly acting as AChE/nicotine anagonist. If so this could be detrimental in terms if curcumine  is used as anticancer agent. The authors should include this effect of curcumine itno the review (for example see the paper of Ximens et al Int. J. Mol. Sci. 2021 vol. 22 p. 972).

Specifically statement in line 310 " ...disturb the activities of specific enzyme preventing NSCLC".  It is not clear preventing NSCLC of what.

This paper is about cell lines, cell signalization, compounds etc. Of course this is complicated even for those who are familiar with abbreviations and terms used in cell biology. For the clarity a list of abbreviations and terms should be included, most properly in the supplementary material. This will make reading of the paper easier for those who are not entirely familiar with plethora of abbreviations used in this particular review. In my opinion here something is missing.

Round 2

Reviewer 1 Report

This version manuscript is ok, in my opinion. Thank you for the implementations.